# Characterisation of Canine and Feline Breast Tumours, Their Metastases, and Corresponding Primary Cell Lines Using LA-REIMS and DESI-MS Imaging

**DOI:** 10.3390/ijms25147752

**Published:** 2024-07-15

**Authors:** Adrienn Molnár, Gabriel Stefan Horkovics-Kováts, Nóra Kucsma, Zsuzsanna Szegő, Boglárka Tauber, Attila Egri, Zoltán Szkupien, Bálint András Deák, James S. McKenzie, Julianna Thuróczy, Richard Schäffer, Gitta Schlosser, Gergely Szakács, Júlia Balog

**Affiliations:** 1Hevesy György PhD School of Chemistry, ELTE Eötvös Loránd University, H-1117 Budapest, Hungary; molnaradrienn97@gmail.com (A.M.); gabriel.horkovics@gmail.com (G.S.H.-K.); 2Waters Research Center, H-1031 Budapest, Hungary; zs.szego@gmail.com (Z.S.); attila.egri@gmail.com (A.E.); richard.schaffer@yahoo.com (R.S.); 3MTA-ELTE Lendület (Momentum) Ion Mobility Mass Spectrometry Research Group, Faculty of Science, Institute of Chemistry, ELTE Eötvös Loránd University, H-1117 Budapest, Hungary; gitta.schlosser@ttk.elte.hu; 4Institute of Molecular Life Sciences, HUN-REN Research Centre for Natural Sciences, H-1117 Budapest, Hungary; kucsmano@gmail.com (N.K.); szakacs.gergely@ttk.hu (G.S.); 5Qamcom Central Europe, H-1052 Budapest, Hungary; boglarka.tauber@2550.engineering; 6Independent Researcher, H-1141 Budapest, Hungary; szkupien.zoli@gmail.com; 7Department of Pathology, Forensic and Insurance Medicine, Semmelweis University, H-1085 Budapest, Hungary; deak.balint.andras@semmelweis.hu; 8Department of Metabolism, Digestion and Reproduction, Faculty of Medicine, Imperial College London, London W12 0NN, UK; j.mckenzie@imperial.ac.uk; 9Animal Health Center Budafok, H-1221 Budapest, Hungary; thuroczy.julianna@gmail.com; 10Center for Cancer Research, Medical University of Vienna, 1090 Vienna, Austria

**Keywords:** cancer research, ambient ionisation, mass spectrometry imaging, metabolic fingerprints, primary cell lines, lipidomics

## Abstract

Breast cancer, a complex disease with a significant prevalence to form metastases, necessitates novel therapeutic strategies to improve treatment outcomes. Here, we present the results of a comparative molecular study of primary breast tumours, their metastases, and the corresponding primary cell lines using Desorption Electrospray Ionisation (DESI) and Laser-Assisted Rapid Evaporative Ionisation Mass Spectrometry (LA-REIMS) imaging. Our results show that ambient ionisation mass spectrometry technology is suitable for rapid characterisation of samples, providing a lipid- and metabolite-rich spectrum within seconds. Our study demonstrates that the lipidomic fingerprint of the primary tumour is not significantly distinguishable from that of its metastasis, in parallel with the similarity observed between their respective primary cell lines. While significant differences were observed between tumours and the corresponding cell lines, distinct lipidomic signatures and several phospholipids such as PA(36:2), PE(36:1), and PE(P-38:4)/PE(O-38:5) for LA-REIMS imaging and PE(P-38:4)/PE(O-38:5), PS(36:1), and PI(38:4) for DESI-MSI were identified in both tumours and cells. We show that the tumours’ characteristics can be found in the corresponding primary cell lines, offering a promising avenue for assessing tumour responsiveness to therapeutic interventions. A comparative analysis by DESI-MSI and LA-REIMS imaging revealed complementary information, demonstrating the utility of LA-REIMS in the molecular imaging of cancer.

## 1. Introduction

Breast cancer is one of the most common cancers in the world and the second leading cause of cancer death among women [1]. Like all cancers, breast cancer is a complex disease characterised by intra- and intertumoral heterogeneity, which have significant impact on prognosis and treatment [2]. In most patients, primary tumours respond to treatment, and the main cause of mortality is linked to the metastases of more aggressive subtypes [3].

One of the most aggressive subtypes of breast cancer is triple-negative breast cancer (TNBC), representing 15% of breast cancer cases, which is associated with a higher risk of metastasis and recurrence. Treatment options are limited to chemotherapy as this tumour type does not express oestrogen, progesterone, or HER2 receptors; so, hormone or targeted therapy is not applicable [4]. Unfortunately, chemotherapy shows limited efficacy and is often accompanied by serious adverse effects and resistance [5]. Therefore, the development of new therapeutic strategies is of major importance [6]. Prognostic markers in primary tumour biopsies and relevant primary cell lines are used to assess the likelihood of metastasis and to predict responses to treatment.

Cell lines are also extensively used as in vitro model systems for cancer research and drug discovery to understand the cellular and molecular biology of cancer [7]. Neve et al. compared the genomic and molecular characteristics of 51 cell lines with primary breast tumours and showed that this panel can be used to identify molecular features that predict response to targeted therapies [8]. While questions have been raised in the past about the representativeness and validity of immortalised cell lines for breast cancer [9], they remained an effective experimental tool with proven clinical benefits [10]. Whereas cell lines provide a reproducible and easily accessible model, in vitro cultured cells are selected for rapid growth, resulting in altered gene expression profiles, signalling pathways and biological functions [11].

Primary cancer cell cultures established from tumours have emerged as a relevant source of information for personalized therapy and drug response prediction [12]. Studies have proved that primary breast cancer cell lines are suitable for pioneering the development of innovative cancer therapies [13]. Further characterisation of current and prospective cell lines as well as studies of therapeutic agents on cell lines should provide a valuable resource for a better understanding of breast cancer.

One of the most important components of cells is the cell membrane, which is rich in various lipid species that play important roles in cell signalling and cell–cell interaction, which are often deregulated in cancer and other diseases [14]. Lipidomic profiles of breast cancer specimens and cell lines were extensively examined [15], showing the utmost importance of understanding lipidomic differences at the molecular level.

Cellular lipid analyses by mass spectrometry-based lipidomics have become a powerful tool/technique in cancer research [15]. Mass spectrometry coupled with chromatographic separation is currently the primary method for the analysis of lipids. Mass spectrometry imaging is also playing an increasingly important role in understanding the spatial localization of tumour and tumour–normal border in tissues. In addition, ambient ionisation mass spectrometry, including Desorption Electrospray Ionisation Mass Spectrometry (DESI-MS), has gained ground over the past decade due to its ability to provide rapid information about samples while requiring minimal or no sample preparation [16,17,18]. This nearly non-destructive method can generate ions from sample surfaces without special sample preparation, labelling, or addition of any matrix. DESI-Mass Spectrometry Imaging (DESI-MSI) has found its way into cancer research and further disease diagnosis application areas due to its ability to distinguish normal and abnormal tissue differences through analyses of lipidomic and metabolic profiles [19,20].

Laser-Assisted Rapid Evaporative Ionisation Mass Spectrometry (LA-REIMS) is another recently developed ambient ionisation technique [21,22], using infrared laser ablation for the evaporation of biological samples to generate an aerosol containing charged particles, which is then subsequently introduced into the atmospheric interface of the mass spectrometer for further analyses. This method, like other laser desorption methods, provides rich molecular profiles of lipids and small metabolites, which enables differentiation between tissue types (e.g., tumour-like and non-cancerous) in the case of tissue measurements and opens new possibilities for cancer research [23,24]. Coupling LA-REIMS to an imaging platform allows precise sampling of tissue sections from microscope slides. Tissue samples can be examined over a range of a few tens of micrometres [25], and even sampling from uneven sample surfaces has already been demonstrated [26]. LA-REIMS is an alternative and complementary technique to DESI-MS, and its application for the analysis of cancer cells is demonstrated in the following study using an LA-REIMS imaging setup built in-house in a laboratory environment.

In this study, we compare the profiles of breast tumours, their metastases, and the cell lines established from both the primary and metastatic tumours, employing the newly developed LA-REIMS imaging platform. Based on the comprehensive molecular information derived from lipidomic patterns, we identify both similarities and differences between the primary tumours and their metastases. Additionally, we demonstrate changes associated with the corresponding primary cell lines.

## 2. Results

A procedure was developed to obtain comprehensive molecular information and to perform comparative molecular studies on tissues obtained from canine and feline tumours, their metastases, and the cell lines established from the same primary and metastatic tumours. A breast simplex tubulopapillar carcinoma (BSTC), its skin metastasis (SM), and a breast adenocarcinoma (BA) and its lung metastasis (LM) were examined. Fresh tumour tissues obtained from veterinary surgeries were cryopreserved for later analyses. In parallel, primary cultures were established from cryopreserved tumour samples. Analyses of the tumour and primary cell line samples by DESI-MS and LA-REIMS imaging allow a comprehensive comparison of tissue specimens and the corresponding primary cells using ambient ionisation technologies (Figure 1).

Canine and feline breast tumours, along with their metastases and the corresponding primary cell lines, established through extended cultivation procedures, were examined at the molecular level by LA-REIMS imaging and benchmarked against DESI-MSI. The results obtained with the two technologies were continuously compared, leading to the compilation of a comprehensive molecular profile for the examined samples. To obtain the maximum amount of information, the following analyses were carried out: (1) First, we analysed the tissue specimens, comparing normal, tumorous, and necrotic (if present) regions identified through pathological annotation. Next, we explored the molecular fingerprints of both the primary and metastatic tumour samples, assessing their similarities and differences. (2) Second, we characterised the cell pellets of immortalised cells. (3) Third, we compared the lipidomic fingerprints of the tumours with those of the respective immortalised cell lines.

### 2.1. Tumour Tissue Measurement Results with LA-REIMS and DESI-MS Imaging Technologies

The tumour samples were characterised by LA-REIMS and DESI-MS imaging, involving the following experimental steps: (1) First, different tissue parts of the sections (normal, tumorous and necrotic, if present) were differentiated by using gold-standard pathological annotation (Figure 2A). (2) As we were interested in the natural clustering of spectra regarding the tissue sections, considering unknown patterns in natural data, we used an unsupervised peak-picking algorithm (k-Nearest Neighbours (kNN) algorithm) (for details, see Section 4) to explore the natural clustering of the tissue and to generate peak lists from each cluster separately, then combining them into a full peak list. These peak lists were then used to determine the main differences in the lipid composition of the different tissue parts (Figure 2B). (3) An image was generated for visualisation of the distribution of specific peaks corresponding to the different parts of the tissue section (Figure 2C). (4) To determine the overlap between gold-standard pathological annotation and the unsupervised clustering method, the results were compared as follows. The pathological annotation was taken as ground truth for tumorous and normal regions (necrotic if present), and the predictive strengths of the unsupervised imaging technologies were evaluated by identifying the true-positive (TP—tumorous ground truth, tumorous predicted), true-negative (TN—normal ground truth, normal predicted), false-positive (FP—normal ground truth, tumorous predicted), and false-negative (FN—tumorous ground truth, normal predicted) regions. The total areas of these regions were then normed by the total area of the section to obtain a confusion matrix containing the relative frequencies of the four classifications. The confusion matrices were evaluated based on their accuracy (ACC) and sensitivity (SEN) (see Section 4) (Figure 2D).

The process of tissue analysis is represented by the BSTC for both DESI-MS and LA-REIMS imaging (Figure 2); the results of all the tissue samples can be found in Appendix A.

(A) The pathological annotation identified normal and tumorous parts in the samples obtained from BSTC (Figure 2A) and its SM (Appendix A); normal, tumorous, and necrotic parts in the samples obtained from BA (Appendix A); and tumorous and necrotic parts in the samples obtained from the LM of BA (Appendix A). (B) The raw imaging data were analysed with the unsupervised algorithm to differentiate each part of the tissue sections (Figure 2B and Appendix A). Peak lists were generated showing the major lipidomic differences between the different parts. (C) Images were generated based on the identified lipidomic differences (Figure 2C and Appendix A). (D) Confusion matrices were obtained, and the results were determined by accuracy (ACC) and sensitivity (SEN). LA-REIMS imaging measurements provided consistent results with over 0.80 sensitivity and over 0.80 accuracy for all tissue specimens (BSTC–SEN = 0.8103, ACC = 0.8034, SM–SEN = 0.8048, ACC = 0.8308, BA–SEN = 0.8094, ACC = 0.8261, LM–SEN = 0.8621, ACC = 0.8151) with 0.86 sensitivity being the highest for LM and 0.83 accuracy for SM (Figure 2D and Appendix A). DESI-MSI measurements resulted in lower accuracy and sensitivity for BSTC (SEN = 0.7014, ACC = 0.6539) and LM (SEN = 0.7154, ACC = 0.6036), little lower for BA (SEN = 0.7801, ACC = 0.7697) and a similar accuracy with 0.7989, and an exceptionally great sensitivity with 0.9397 for SM (Figure 2D and Appendix A). According to our findings, the identified area of the tumour using unsupervised kNN clustering of MS imaging spectra is slightly larger compared to the pathological boundaries, suggesting that chemical imaging might be able to identify tumour-related alterations, which are not represented in the morphology analysed by conventional methods. However, these hypotheses need to be validated. 

#### 2.1.1. LA-REIMS Imaging Measurements

Tissue section imaging measurements provided unique metabolic fingerprints. Normal, tumorous, and necrotic parts of the tumour specimens could be clearly distinguished by LA-REIMS imaging. Results of the imaging measurements were consistent with the pathological observations (Figure 2, Appendix A).

The next step was to identify characteristics at the molecular level in the tumour specimens. After successfully distinguishing the different parts within the tissues, we proceeded to analyse the main lipidomic differences between the normal and tumorous parts (and necrotic if present) in the tissue specimen to gain a deeper understanding of tumour characteristics.

Adjacent normal (non-cancerous) tissue regions identified in the BSTC, its SM, and BA tissue specimens were compared to the respective tumorous regions. All tissues revealed major differences between cancerous and non-cancerous regions. Different types of ceramides (Cer), including ceramide phosphates (CerP), phosphatidylethanolamine-ceramides (CerPE), and hexosyl ceramides (HexCer); as well as phosphatidylethanolamines (PE), phosphatidic acids (PA), and characteristic phosphatidylinositols (PI) were identified. Ceramides were more abundant in non-cancerous regions, while PEs, PAs, and PIs were more abundant in tumorous regions. These findings were best represented in the BA sample, where major differences between cancerous and adjacent healthy tissue included ceramides such as HexCer(30:1;O2), CerPE(32:1;O5), CerP(38:5;O3), CerPE(36:1;O2), and Cer(42:5;O3); PEs such as PE(O-36:5)/PE(P-36:4), PE(O-38:5)/PE(P-38:4), PE(O-38:4)/PE(P-38:3), and PE(38:4); and a characteristic PI (PI(38:4)) (Figure 3, Appendix A).

In the case of BSTC and its SM, major lipidomic differences between cancerous and normal tissue parts consisted of more abundant PAs and PEs, and less ceramides. Major lipidomic differences were in Cer(36:1;O2), and PA(36:3), PA(36:2), PE(P-34:2), PE(O-34:2)/PE(P-34:1), PE(O-36:5)/PE(P-36:4), and PE(O-38:5)/PE(P-38:4) as well as PE(34:2), PE(34:1), PE(36:3), PE(36:2), and PE(36:1) (Appendix A).

Interestingly, significant differences between the tumorous and necrotic regions were observed particularly in PEs and ceramides. Ceramides were found to be more abundant in the necrotic regions, highlighting an alteration in PEs in both the tumorous and necrotic regions. Ether-linked phosphatidylethanolamines (PEs) emerged as a major distinction between the two regions. In addition to the previously identified PE(O-36:5)/PE(P-36:4), PE(O-38:5)/PE(P-38:4), and PE(O-38:4)/PE(P-38:3), PE(O-36:3)/PE(P-36:2), PE(O-38:6)/PE(P-38:5), and PE(O-38:3)/PE(P-38:2), as well as PE(36:2), PE(38:4), PE(38:3) and several ceramides including HexCer(30:1;O2), Cer(42:2;O2), CerPE(36:1;O2), HexCer(36:2;O6), and CerPE(44:2;O2) were identified as significant differences (Appendix A).

The BA tissue provided an opportunity to investigate differences between normal and necrotic regions. The major difference in molecular composition between the two regions was observed in the abundance of PEs and ceramides. Non-cancerous regions were more abundant in ceramides, while necrotic regions were rich in ether-linked PEs. The previously observed series of ether-linked PEs remained significant with PE(O-38:6)/PE(P-38:5), PE(O-38:5)/PE(P-38:4), PE(O-38:4)/PE(P-38:3), and PE(O-38:3)/PE(P-38:2) being notably more abundant in the necrotic region (Appendix A).

Next, we focused on the relationships between primary tumours and their metastases. PCA analysis of the section measurements showed a high degree of similarity between the primary tumours and their metastases (Figure 4A,C). In both cases, the primary tumour and its metastasis were located closer to each other in the PCA space than to the adjacent and necrotic parts of the tissue sections. A PCA space distance of 1.542 was observed for the BSTC and its SM (Figure 4B), while a PCA space distance of 1.025 was observed for the BA and its LM (Figure 4D). Meanwhile, the relative PCA distance tables revealed significant differences among the distinguished parts of the tissues (PCA space distance over 2.2) (Figure 4B,D).

Our observations show that the lipidomic patterns of the primary tumour and the metastasis do not differ significantly; however, changes in ceramides, PAs, and PEs indicate alterations in the tumour lipid metabolism compared to the adjacent tissue. Three lipids, PA(36:3), PA(36:2), and PE(36:2), appeared in both cases, indicating their role in metabolism changes. In addition, lipid series of which PE(38:4) and PE(38:3); PE(34:2) and PE(34:1); or PE(36:3), PE(36:2), and PE(36:1) are components may also play a role in the changes.

#### 2.1.2. DESI-MSI Measurements

DESI-MSI was used as a complementary, benchmark technology to image the tumours and cells. As expected, the DESI-MSI measurements also resulted in rich lipidomic fingerprints. The results aligned with the LA-REIMS imaging results and were consistent with the previously described pathological observations. The normal, tumorous, and necrotic parts of the tumour specimens could be differentiated by DESI-MSI, yielding similar results to those previously described with LA-REIMS imaging (Figure 2, Appendix A).

After assigning the imaging measurements with the appropriate tumour tissue regions, their lipid profiles were investigated as it was presented in the case of LA-REIMS imaging measurements to identify molecular characteristics. The results revealed major differences in ceramides (Cer), phosphatidylethanolamines (PE), phosphatidylserines (PS), and phosphatidylinositols (PI) between the tumour tissue regions. Compared to the LA-REIMS imaging results, lower amounts of ceramides and no phosphatidic acids were identified, while characteristic PSs and PIs appeared as major differences. Additionally, the identified PEs, PSs, and PIs were more abundant in the tumorous regions.

Adjacent normal and cancerous tissue regions were compared. The BA tissue revealed that the major molecular differences representing the tumorous tissue, identified by DESI-MSI, included the previously identified PEs such as PE(O-36:5)/PE(P-36:4), PE(O-38:5)/PE(P-38:4), PE(O-38:4)/PE(P-38:3), PE(38:4), and PI(38:4). Additionally, PE(O-38:6)/PE(P-38:5), PS(36:1), PS(38:3), and PI(38:3) were the most characteristic of the tumorous regions (Appendix A).

In the BSTC and its SM, distinctive lipids characteristic of the tumours were identified. Notably, a series of previously recognised PEs—PE(O-34:2)/PE(P-34:1), PE(34:1), PE(36:3), PE(36:2), and PE(36:1)—emerged as major molecular differences in the DESI-MSI analysis. Additionally, a series of ether-linked PEs were observed: PE(P-34:2), PE(O-36:3)/PE(P-36:2) and PE(O-36:2)/PE(P-36:1); and PE(O-38:6)/PE(P-38:5), PE(O-38:5)/PE(P-38:4) and PE(O-38:4)/PE(P-38:3); and the characteristic PS(36:1) and PIs including PI(34:2), PI(36:2), and PI(38:4) were also identified (Appendix A).

Comparison of the normal and necrotic regions in the BA tissue revealed the aforementioned ether-linked PEs, along with the characteristic PS and PIs as main differences (Appendix A). These findings mirrored the major molecular differences between the tumorous and necrotic regions together with alterations in PE(36:1), PE(38:4), and PI(38:2) (Appendix A).

PCA analysis of the section measurements revealed a significant similarity between the primary tumour and its metastasis (Figure 4E,G). A PCA space distance of 0.938 was determined for the BSTC and its SM (Figure 4F), and 0.662 was detected for the BA and its LM (Figure 4H), while a PCA space distance of over 1.4 was observed among other parts of the tissues (Figure 4F,H).

The tumorous parts of the primary tumour and metastasis tissue sections were compared using DESI-MSI. Similar to LA-REIMS imaging, the conclusions drawn is that the lipidomic patterns between the two do not differ significantly. Notable alterations were observed in the ether-linked PEs, PSs, and PIs.

#### 2.1.3. Direct Comparison of Tissue Types with DESI-MSI and LA-REIMS Imaging Technologies

Moreover, we aimed to obtain an overview of the measurement results for the tumour tissue specimens from the two ambient technologies and prove the potential of LA-REIMS imaging. Therefore, we conducted a collective analysis of the findings and examined the results to demonstrate that a technology currently under development provides results of similar quality to an already acknowledged, widely used one (Figure 5).

The results demonstrated that LA-REIMS imaging yielded similar results to DESI-MSI in that differentiation among various parts of the tumour tissue sections was achievable. While molecular fingerprints showed clear differences between DESI-MSI and LA-REIMS imaging, the PCA/LDA results did not reflect these distinctions. Clear similarities were observed between the primary tumours and their metastases, as well as among the three adjacent tissue parts and the two necrotic tissue regions. These observations were confirmed by the PCA cross class distance tables (Figure 4).

To complete the benchmark study, we examined how well primary tumours and their metastases resemble each other/can be classified. First, PCA/LDA models were built from the primary tumours, and the metastatic tissue was classified using this model. Next, PCA/LDA models were built from the two metastases and were used for the classification of the two primary tumours. A total of 5 PCA components (a total of 5 to 50 PCA components were tested and optimized) were used to calculate 3 LDA components. If the distance of a newly classified spectrum was greater from the training set class average than 15 times the standard deviation of the full class, the spectrum was labelled ‘outlier’. The results demonstrate the connection between the metastasis and the primary tumour with both technologies and the applicability of LA-REIMS for such purposes. The primary tumour can clearly be identified based on the patterns detected in the respective metastasis with both technologies with 100% accuracy (Appendix A). Identifying the metastasis based on the fingerprints of the primary tumour is not as straightforward as performing this vice versa. In the case of the DESI-MSI measurements, metastatic tumorous regions can be identified with 100% accuracy (Appendix A). In the case of the LA-REIMS imaging measurements, BA’s metastasis can be identified with 100% accuracy; however, the identification of the BSTC’s metastasis is not unambiguous with 83.59% accuracy; some misidentifications are observed. The other primary tumour and its metastasis are also assigned (Appendix A).

### 2.2. Cell Pellet Measurement Results of the Immortalised Cells with DESI-MSI and LA-REIMS Imaging Technologies

In the next step, we obtained a comprehensive molecular profile for the immortalised primary cells established from the previously characterised tumours. Cell lines were established from the above-mentioned primary tumours and their metastases (BSTC and its SM and BA and its LM) through several months of cell cultivation. Cells were frozen at every passage. The samples corresponding to immortalised cells were dried on slides and measured by LA-REIMS imaging and DESI-MSI. The analysis of cell pellet measurements yielded rich molecular profiles, where we again mainly focused on the lipidomic region (600–900 *m*/*z* range). The mass spectra of cells from different origins revealed notable differences in their metabolic fingerprints (Appendix A).

For LA-REIMS imaging measurements, multiple series of PAs and PEs were identified, and the most abundant were listed (Appendix A). Among them, PA(34:1), PA(36:2), PA(36:1), PA(38:3), PA(38:2), PE(36:1), PE(O-38:6)/PE(P-38:5), PE(O-38:5)/PE(P-38:4), and PE(38:4) were characteristic of all four cell lines. Moreover, PE(36:1), PE(O-38:6)/PE(P-38:5), and PE(O-38:5)/PE(P-38:4) were among the most abundant peaks in the DESI-MSI measurements as well. However, in the case of DESI-MSI, other PSs and PIs including PS(36:1), PI(38:4), and PI(38:3) were identified for all cell lines (Appendix A).

Major differences between the two techniques indicated that in the case of LA-REIMS imaging, mass spectra (600–900 *m*/*z* region) are more abundant in the lower mass region containing more ceramides, PAs, and PEs. In the case of DESI-MSI, mass spectra are more abundant in the upper mass region, containing more phosphatidylglycerols (PGs), PIs, and PSs. The presence of PAs is highly characteristic of LA-REIMS imaging, while the presence of PGs, PSs, and PIs is characteristic of DESI-MSI.

According to the PCA/LDA analysis and the mass spectra, the immortalised cells of the primary tumour and its corresponding metastasis revealed great similarities with both technologies. Meanwhile, cells of different origins were clearly distinguishable from each other, and differences regarding the ambient techniques were also observed. PC1 clearly differentiated the spectra acquired by LA-REIMS and DESI-MS imaging, while PC2 distinguished the two different tumour types (Figure 6). There is no clear separation between the primary and metastatic cell lines.

### 2.3. Cell Line–Tumour Tissue Comparison with LA-REIMS Imaging and DESI-MSI

The next aim of this study was to compare the tumours with their corresponding primary cell lines to explore their similarity and sensitivity profiles. The objective was to identify characteristic peaks and make an attempt to discover specific marker molecules characterising both the tumour and the respective cancer cell line. Mass spectra of the homogenous cell cultures were compared with those of the tumorous parts of tissues, revealing significant differences. Typical mass spectra for LA-REIMS imaging and for DESI-MSI are shown in Figure 7 and Appendix A, respectively.

Since PCA relative distances and the cross-validation results demonstrated the similarity of tumour–metastasis pairs and the similarity of their cell lines, we examined differences between the two by directly comparing cancerous tissue spectra to the spectra obtained for the respective established cell line. For the LA-REIMS imaging measurements, the PAs and PEs are responsible for the major differences between the tissues and cells. The results showed that the PEs are more abundant in the tissue samples while the PAs are more abundant and significantly characteristic to the primary cells. Major differences were identified and five PAs as PA(34:1), PA(36:2), PA(36:1), PA(38:3), and PA(38:2) were found to be mainly responsible for the observed differences. These five lipids correspond to the characteristic PAs of all the cell lines identified previously. Meanwhile, the PEs responsible for the differences, as PE(O-38:5)/PE(P-38:4) and PE(38:4) for the BA tissues and PE(34:2), PE(36:3), PE(36:2), and PE(36:1) for the BSTC tissues were previously shown to be characteristic of the tumorous tissue regions.

The DESI-MSI measurements also revealed several differences between tumour tissues and primary cells (Appendix A). Since no or very few PAs were identified, the results did not give as clearly distinguishable differences as in the case of LA-REIMS imaging. However, results also showed, similar to those observed for LA-REIMS imaging, that in almost all cases, PEs such as PE(O-38:5)/PE(P-38:4) and PE(38:4) were more characteristic of the tumour tissue regions, while PSs and PIs were more characteristic of primary cells. Taken together, PIs such as PI(38:4), PI(38:3), and PI(38:2) are characteristic of DESI-MSI tissue (tumorous tissue region) and cell measurements, respectively.

Despite the significant differences observed between tumour and cells, notable similarities could also be detected. Differences were identified by a supervised classification method examining relative intensities of the peaks, while we examined the similarities evaluating the mass spectra manually focusing on the peaks regardless of their intensities. We sought to reinforce the point that cell lines are a relevant source of information for personalised therapy and drug response predictions. In the case of the LA-REIMS imaging measurements, all the tumour samples and their respective cell lines revealed three major similarities including PA(36:2), PE(36:1), and PE(O-38:5)/PE(P-38:4), completed with PE(O-38:6)/PE(P-38:5) for BA and its LM, and PE(38:4) for BSTC and its SM. Additionally, several other similarities such as PA(34:1), PE(O-38:4)/PE(P-38:3), PE(36:2), PA(36:1), and PA(38:3) were also observed (Appendix A). Similarly, the DESI-MSI measurements also revealed three major similarities between all tumour specimens and cells, including PE(O-38:5)/PE(P-38:4), PS(36:1), and PI(38:4). Further similarities included PE(O-38:6)/PE(P-38:5) and PI(38:3) for BA and its LM and PE(36:1) and PE(38:4) for BSTC and its SM, as observed with LA-REIMS imaging measurements. Additionally, other similar molecules such as PS(38:3) were also observed (Appendix A).

## 3. Discussion

A comprehensive molecular study was performed on canine and feline breast tumours, their metastases, and the corresponding immortalised primary cell lines by two ambient technologies: Desorption Electrospray Ionisation Mass Spectrometry Imaging (DESI-MSI) and Laser-Assisted Rapid Evaporative Ionisation Mass Spectrometry (LA-REIMS) imaging. In this work, we examined tumour tissue sections at the molecular level, identified tumour characteristic lipids, proved the connection between the primary tumours and their metastases, examined tumour and cell line relations, and compared the two technologies. We demonstrated that LA-REIMS imaging is suitable for the rapid analysis of tissues and cells. Differences between the tumour tissue regions (adjacent, tumorous, and necrotic) were determined and marker lipids specific to the tumour regions were identified. Unsupervised kNN clusters of DESI-MS and LA-REIMS imaging tissue spectra were compared to gold-standard H&E-stained pathological examination, and our findings suggested a higher tumour presence and more detailed complexity within the tumour compared to the pathological annotation. Our overall accuracy and sensitivity were as high as >80% with LA-REIMS and >70% with DESI-MS imaging, even though we were using an unsupervised clustering method. Unsupervised methods provide an insight to the molecular fingerprint characteristics of the classes without the danger of overfitting the data, and as our goal was to detect molecular patterns within the tissues and cells, not to estimate the classification accuracy of a pre-built database, we found in our case that unsupervised methods such as PCA and kNN are more suitable compared to supervised classification algorithms. While conventional histopathology is based on staining and morphology results, MSI is based on chemical information, which can provide more detailed insights to the actual cell status. Moreover, the necessary time to have detailed identification with MSI is a few hours compared to the 1–2 days necessary for conventional histopathology results. It is well known that phospholipid alterations in the cell membrane are linked to various human diseases including cancer [27]; thus, our main goal was to investigate a rapid, on-line monitoring tool for the analysis of cell membrane lipids and, through that, the cancerous state of cells.

DESI-MSI and LA-REIMS imaging operate on different principles. While DESI uses a charged spray to physically remove molecules from the surface of the samples [17], LA-REIMS is based on rapid heating of the sample and thus mobilizing the particles through thermal ‘explosion’ [28]. An interesting phenomenon observed during rapid thermal energy conduction on the sample is the thermal-based ammonia loss of PEs, resulting in an [M-NH_4_]^−^ ion in contrast to the usual [M-H]^−^ molecules. This observation aligns with our previous findings [25]. Consequently, the obtained mass spectra are also different, with LA-REIMS imaging giving more abundant mass spectra mainly in the lower mass region of the studied 600–900 *m*/*z* range and with DESI-MSI yielding more abundant spectra in the upper mass region. By investigating molecular coverage, our aim was to characterise the molecular fingerprint of the samples, to investigate the extent to which the two technologies provide complementary information about the samples, and to search for tumour-specific lipid marker molecules. While we recognise further potential characterising smaller molecules, this was not the scope of this study.

We identified the most abundant molecules characterising the tumorous tissue regions with both technologies. The results showed that there is a significant overlap between the lipids identified by DESI-MSI and LA-REIMS imaging but also that each technology provides significant complementary information. Although several PAs, PEs, PSs, and PIs characterise the samples, the overlap in the molecules is significant regarding only PEs, given that PAs were not abundant in DESI-MSI, while PSs and PIs were not abundant (except PI(38:4)) in the spectra acquired with LA-REIMS imaging. On this basis, we can conclude that a comprehensive and partially complementary molecular information of the samples can be obtained with the two techniques. Although the relative quantity of the molecules of interest identified by both technologies are different, the studied biological alteration appears in both ionisation techniques, providing consistent results. Hence, the identification of specific tumour markers with one technique implies that they can be subsequently sought using another technique targeting different tumour regions.

According to the cross-validation results, the primary tumour can clearly be identified based on the patterns detected in the respective metastasis with 100% accuracy, allowing the characterisation of the primary tumour by examining the metastasis. Our observations suggest that, at least in the studied examples, the lipidomic patterns of the primary tumour and metastasis do not differ significantly. The major molecules identified within the tumour–metastasis pairs are nearly identical in the case of DESI-MSI, and there is a significant similarity in the case of LA-REIMS imaging. Variations in ceramides, PAs, and PEs indicate an alteration in the tumour lipid metabolism. Detailed analyses of the tumour tissue regions identified three highly abundant peaks corresponding to PE(O-36:5)/PE(P-36:4), PE(O-38:5)/PE(P-38:4), and PE(38:4), indicating the potential of these lipids as possible tumour markers. In agreement with our findings, it was shown in the 1980s that there is an enrichment of poly-unsaturated fatty acid-containing phospholipids and PEs in general in breast cancer [29]. Using DESI-MSI, Porcari et al. identified PE(38:4) and PE(O-38:5) as breast tissue biomarkers. In addition to these, our study also identified several other biomarker lipids reported by them, including PS(38:4), PS(40:4), PI(36:2), PI(38:3), PE(36:2), PE(O-38:6), PS(36:2), PS(36:1), PI(38:4), and PE(O-38:4) [30,31]. In light of these results, the importance of the characteristic lipids identified in the tumour regions in our measurements may be worth considering and investigating in the study of tumour markers.

The most abundant lipids characterising cells were identified as well. As expected, more PIs and PSs were characterised with DESI-MSI, and a higher number of PAs and PEs were characterised with LA-REIMS imaging. However, since the LA-REIMS signal consisted mainly of PAs, and not a significant number of PAs were identified with DESI-MS, the overlap between the two technologies is slightly lower than that observed for the tumorous tissue regions.

Nevertheless, cell lines showed significantly different characteristics compared to their respective parental tumour tissues (Figure 7 and Appendix A). Differences between the tumorous regions and cells may be explained by several factors. Both the DESI-MSI and LA-REIMS imaging samples were acquired using a 50 µm spot size sprayer or laser; thus, the signal was obtained from multiple cells. In the case of the homogenous cell cultures containing only cancerous cells, this resulted in the analysis of similar cells. However, the tumour tissue contains a variety of different cell types including stromal cells and adjacent normal tissue, resulting in a heterogenous, mixed signal. Moreover, cells change during the long-lasting cultivation procedure as they adapt to the new environment, which is accompanied by the alteration of cell metabolism and the composition of the cell membranes [8].

The similarities and differences between breast cancer tumours and the corresponding cell lines were extensively examined [32]. In our study, we identify characteristic lipid signatures across cell lines and tumours. Of these, PE(36:2), PE(38:4), PS(36:1), and PI(38:4) have been previously shown to be associated with a high metastatic potential [33]. The observed molecular similarities between tumours and their corresponding cell lines support the utilization of cell lines for characterising the phenotype of tumours, including drug sensitivity and metastasis formation.

In summary, our study demonstrates that ambient ionisation mass spectrometry technologies are effective for rapidly monitoring the metabolic state of cells, offering a lipid- and metabolite-rich spectrum within seconds. We demonstrate that immortalised cell lines can be generated from recurrent or metastatic tumours, and our findings establish a strong connection between the cell line and its parental tumour tissue. The workflow relying on ambient ionisation technologies can contribute to the characterisation of in vitro and in vivo models used to assess responses to established or experimental therapies.

## 4. Materials and Methods

### 4.1. Tumour Tissue Handling

Spontaneous breast tumour–tumour metastasis pairs were collected during veterinary surgeries: a feline breast simplex tubulopapillar carcinoma and its skin metastasis and a canine breast adenocarcinoma and its lung metastasis (Table 1). Tumour diagnoses were established by pathological examinations. Required ethical permissions were obtained for animal sample collection (PE/EA/00085-2/2023). Pet owners signed written consent to allow the use of the tumour tissues for further research purposes.

Removed tumour specimens were cut into 1–2 mm pieces, placed in 10% (*V*/*V*) dimethyl sulfoxide (DMSO) (Thermo Fischer Scientific, Waltham, MA, USA) and 90% (*V*/*V*) fetal bovine serum (FBS) (Thermo Fischer Scientific, Waltham, MA, USA) solution, were kept at −20 °C for 3 h, and were stored in liquid nitrogen until further study. In parallel, tumour tissues were frozen at −80 °C until further analysis.

### 4.2. Cell Line Establishment

Cryopreserved tumour samples were thawed and digested in 0.8 mg/mL dispase and 0.54 mg/mL collagenase (Gibco, Thermo Fischer Scientific, Waltham, MA, USA) in supplemented culture medium. The digested tumour suspensions were filtered through a 70 µm filter, and cell cultures were started [34]. The cells were cultured in Dulbecco’s Modified Eagle’s Medium/Nutrient Mixture F-12 (DMEM/F-12) (Gibco, Thermo Fischer Scientific, Waltham, MA, USA) supplemented with 20% (*V*/*V*) fetal bovine serum, 2 mM L-glutamine, 100 units/mL penicillin, and 100 µg/mL streptomycin (Life Technologies, Carlsbad, CA, USA). The cells were cultured at 37 °C, 5% CO_2_, and 20% oxygen through 20 passages, until the cells reached their immortalised forms and immortalised cell lines were established. The cell samples were frozen during each passage. The cell pellets were dried on slides for imaging.

### 4.3. Histology

Frozen tumour tissues were sectioned using a cryotome, and a series of 10 µm thick tissues were mounted on standard glass slides. A section from each was hematoxylin-eosin-stained, and normal, cancerous, and necrotic parts were identified by pathological annotation.

All the samples were measured without any additional sample preparation steps.

### 4.4. Experimental Design/Instrumentation

#### 4.4.1. Measurements by LA-REIMS Imaging

The samples were characterised by the LA-REIMS imaging platform (for research use only) coupled to a Xevo^TM^ G2-XS Q-TOF MS (Waters Corporation, Milford, MA, USA). For tissue ablation, a commercially available Opolette HE 2940 Optical Parametric Oscillator (Opotek LLC, Carlsbad, CA, USA) laser was used at a 2940 nm wavelength. The maximum laser energy output was used, which was further attenuated with an optical iris diaphragm (CP20S—30 mm cage system), a CaF_2_ plate beamsplitter (BSW511), and a neutral density filter (NDIR10A) (Thorlabs Inc., Newton, NJ, USA). To achieve the best energy distribution, the setup was equipped with a ZnSe aspheric lens (12.7 mm focal length) (Thorlabs Inc., Newton, NJ, USA) [25]. The tumour samples were measured using a spatial resolution of 70 µm, while the cell samples were measured using 100 µm.

A 1 m PTFE tube with an inner diameter of 1.7 mm was used to aspirate the generated aerosol above the sample via a custom-built Direct REIMS interface (Waters Research Center, Budapest, Hungary) [21]. The ablated sample was mixed with propan-2-ol (Merck, Darmstadt, Germany) containing 0.1 ng/µL leucine enkephalin (Thermo Fischer Scientific, Waltham, MA, USA) that was constantly injected in front of the MS inlet capillary with a 150 µL/min flow rate [28].

#### 4.4.2. Measurements by DESI-MSI

The samples were also characterised by Desorption Electrospray Ionisation Mass Spectrometry Imaging (DESI-MSI) (for research use only) fitted on a Xevo^TM^ G2-XS Q-TOF MS (Waters Corporation, Milford, MA, USA).

The tumour sections were measured using a spatial resolution of 50 µm, while the cell samples were measured using 100 µm. The height distance between the DESI sprayer and the sample surface was 1.5 mm, the distance between the sprayer and the inlet capillary was 4 mm, the scanning speed of the DESI sprayer was 0.5 and 1.0 mm/sec, respectively, the spray angle was 70°, the spray voltage was 0.7 kV, the inlet capillary temperature was 500 °C, the nitrogen gas pressure was 6–7 psi, and the methanol–water 95:5 solvent containing 0.1 ng/µL leucine enkephalin as standard was used at a 2 µL/min flow rate.

### 4.5. Data Analysis

Data were acquired in negative ion mode, 0.1 s scan time, and a mass-to-charge range of 50–1200 and were processed with our software Abstract Model Builder (AMX, [Beta] version 1.0.58.0, Waters Research Center, Budapest, Hungary) built in-house using multivariate statistics including Principal Component Analysis (PCA), followed by Linear Discriminant Analysis (LDA). The data were visualized using MassLynx^TM^ V4.2 (Waters Corporation, Wilmslow, UK) and were evaluated manually using the top 100 most abundant peaks. Molecules were identified by using exact mass measurements, MS/MS measurements (Appendix A), and the lipidmaps.org database [35].

The imaging measurements were carried out using our custom-built Laris Imager software (Build #188) (Waters Research Center, Budapest, Hungary), and the images were processed with the High Definition Imaging Software 1.6 (Waters Corporation, Milford, MA, USA).

The imaging data were processed with a semi-automatic algorithm producing a clustering and an ordered list of important peaks. The algorithm processes binned data by generating a pairwise distance matrix. Utilizing this matrix, clustering is achieved through the minibatch K-means method, which requires a user-specified number of classes. To identify significant peaks, the following procedure is implemented: A random forest classifier is trained on the peaks, with the target being a binary label distinguishing one of the previous clusters from all others. The classifier employs a shallow boosting tree comprising 200 estimators. Feature importances are extracted using the SHAP Python module, yielding a list of peaks with high usability. This process is repeated for each cluster, and the results are merged to compile a comprehensive list of important peaks. The entire algorithm is implemented in Python 3.8.

The clusters obtained by the kNN algorithm, indicating non-cancerous and tumorous (and necrotic if present) regions, were compared to the pathological annotation via an image processing script written in MATLAB. Since the clustered images and the annotated references were obtained from different sources, their image sizes, resolutions, and orientations were different. To compare them, the annotated reference image was first transformed to match the size and orientation of the clustered image by selecting pairs of corresponding reference points on both images and estimating the best affine transformation between them. The two images, now exactly the same size and orientation, were then compared pixel by pixel, checking whether each pixel was marked as a tumorous region (positive) on the reference annotation and the clustered image. The pixels were classified as true positive (TP), true negative (TN), false positive (FP) or false negative (FN) depending on whether the clustering algorithm correctly predicted tumorous or non-cancerous tissue (necrotic in the case of the BA’s LM tissue) as identified by the pathological annotation, taken as ground truth. (Only pixels identified as part of the sample on both images were classified.) The total areas of the four classes were then normed by the total area of the tissue section, and a confusion matrix was calculated for each sample containing the relative frequencies of these classes in the sample. The performance of the algorithm was characterised by its accuracy (ACC = (TP + TN)/(TP + TN + FP + FN))—ratio of correctly classified pixels to all pixels and sensitivity (SEN = TP/(TP + FN))—ratio of true positives to all positives for each sample.

To determine the similarity between the different tissue parts (normal, tumorous, and necrotic) in the PCA/LDA space, PCA space distances using Euclidean distances were used. The peaks giving the most important differences were determined by a supervised classification method, Linear Support Vector Classification (LSVC). An *F*-test was applied with a significance of 0.05 to test if the variance of the two samples was the same. If the variances differed, a Welch test was applied to determine the significance of the means differing (*p*-value); if the variances were found to be the same, a *t*-test was used instead. Fold change values were also calculated: the average intensity in a given bin of spectra in the tumour class was divided by the average intensity in a given bin of spectra in the other (adjacent) class, and the result was rounded to two decimal places. This means that the fold change value is greater than one per bin if the intensity average of the tumour class is greater than that of the other class; otherwise, it is less than one per bin.

## Figures and Tables

**Figure 1 ijms-25-07752-f001:**
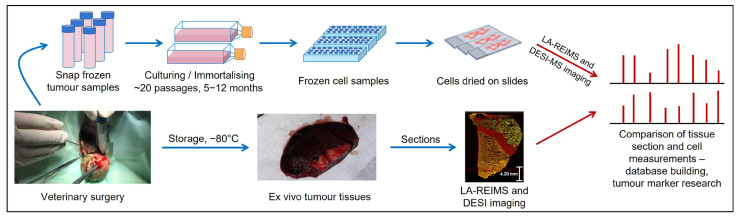
Pipeline of the analysis of cryopreserved samples by mass spectrometry imaging.

**Figure 2 ijms-25-07752-f002:**
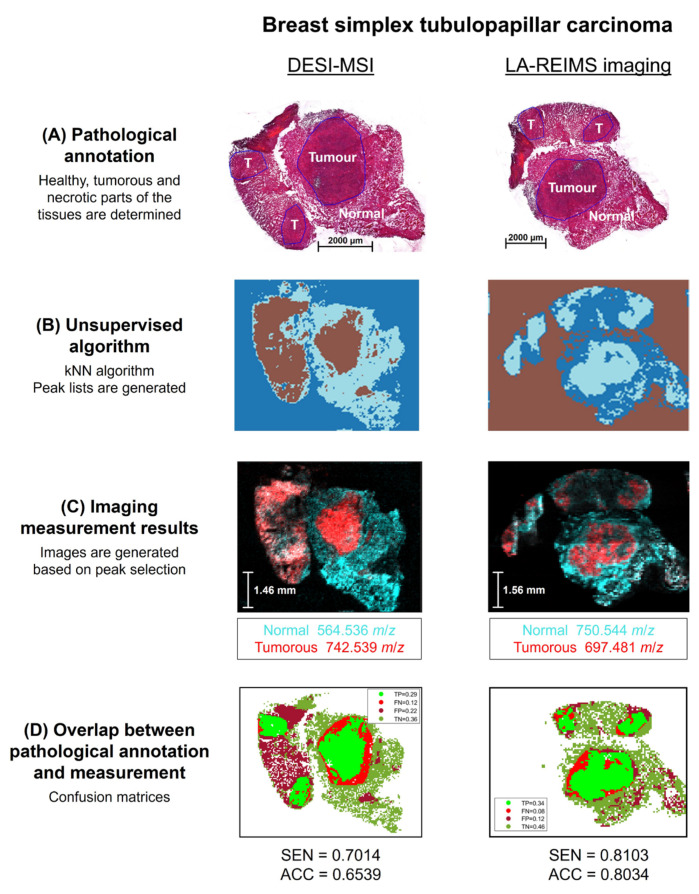
Analysis of adjacent BSTC tissue sections with DESI-MS and LA-REIMS imaging. (**A**) Normal and tumorous (T) tissue parts were identified by pathological annotation. (**B**) kNN algorithm was used to identify clusters and the associated peak lists. (**C**) Images were generated based on the peak lists, selecting *m*/*z* 564.536 (Cer(36:1;O2)) and *m*/*z* 742.539 (PE(36:2)) for DESI-MSI and *m*/*z* 750.544 (PE(O-38:5)/PE(P-38:4)) and *m*/*z* 697.481 (PA(36:3)) for LA-REIMS imaging. (**D**) Confusion matrices containing true-positive (TP), true-negative (TN), false-positive (FP) and false-negative (FN) classifications were obtained. Overlap between the pathological annotation and the imaging measurements resulted in 0.7014 sensitivity and 0.6539 accuracy for DESI-MSI and 0.8103 sensitivity and 0.8034 accuracy for LA-REIMS imaging.

**Figure 3 ijms-25-07752-f003:**
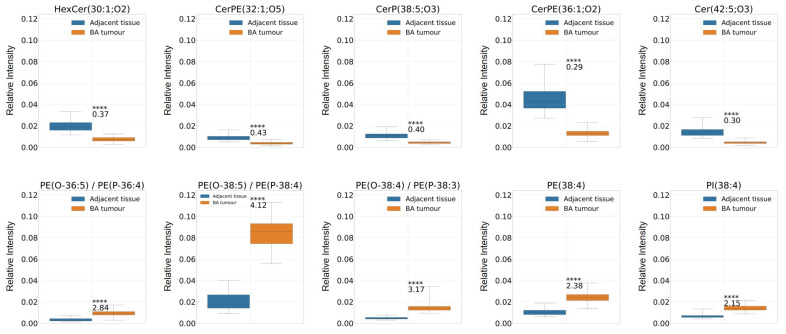
Major lipidomic differences between cancerous and non-cancerous parts of the BA tissue sample (tumour, n=166; adjacent tissue, n=118, where n= number of analysed spectra) with a significance level **** indicating *p* < 0.0001 and fold change values (with >1 indicating greater intensity average of the tumour class and <1 indicating greater intensity average of the adjacent class). Ceramides (HexCer(30:1;O2), CerPE(32:1;O5), CerP(38:5;O3), CerPE(36:1;O2), and Cer(42:5;O3)) are more abundant in normal/adjacent tissue regions while phospholipids (PE(O-36:5)/PE(P-36:4), PE(O-38:5)/PE(P-38:4), PE(O-38:4)/PE(P-38:3), PE(38:4), and PI(38:4)) are more abundant in tumorous regions.

**Figure 4 ijms-25-07752-f004:**
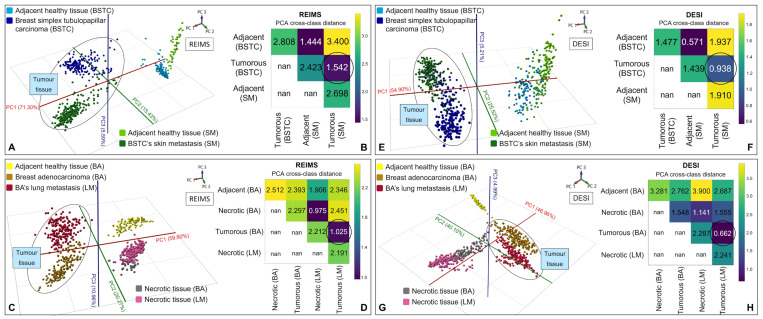
LA-REIMS imaging measurements: (**A**) PCA analysis of normal (adjacent) and tumorous parts (marked by a circle) of the BSTC and its SM tissue sections and (**B**) their PCA relative distance table regarding the different tissue parts; (**C**) PCA analysis of the normal (adjacent), tumorous, and necrotic parts of the BA and its LM tissue sections and (**D**) their PCA relative distance table regarding the different tissue parts. DESI-MSI measurements: (**E**) PCA analysis of the BSTC and its SM and (**F**) their PCA relative distance table; (**G**) PCA analysis of the BA and its LM and (**H**) their PCA relative distance table.

**Figure 5 ijms-25-07752-f005:**
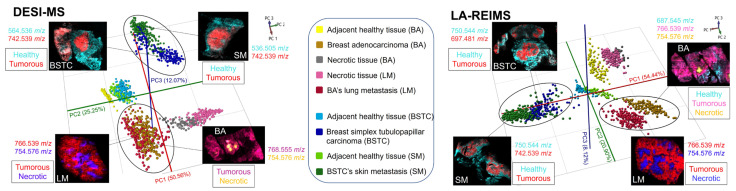
PCA/LDA analysis of normal, tumorous, and necrotic parts of primary tumours and their metastases. The DESI-MSI and LA-REIMS imaging results are presented, and the tumorous tissue parts are linked with the PCA/LDA analysis results. For scale bars see Figure 2 and Appendix A, respectively.

**Figure 6 ijms-25-07752-f006:**
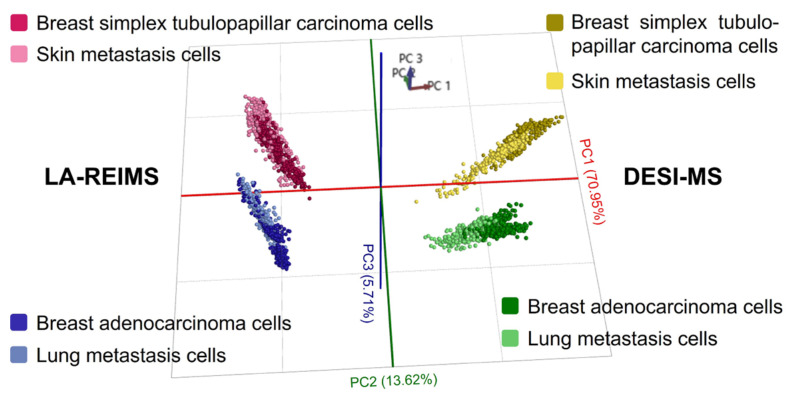
PCA/LDA analysis of the immortalised cells derived from the primary tumours and their metastases measured by LA-REIMS imaging and DESI-MSI.

**Figure 7 ijms-25-07752-f007:**
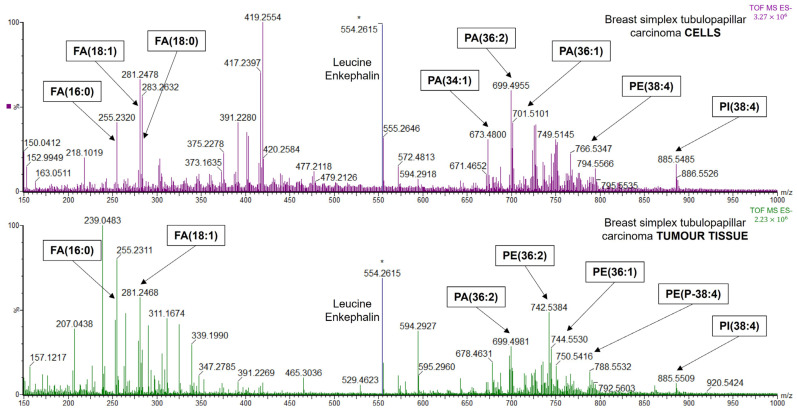
Mass spectra of the primary cell line derived from BSTC and the original tumour tissue (average of 30 scans) measured by LA-REIMS imaging. There is a rich and distinct lipidomic signal in the 600–900 *m*/*z* region, while the metabolic fingerprint of smaller molecules, including fatty acids (FA), is significantly different in the 250–350 *m*/*z* region. Mass spectra were lock mass corrected to the internal standard leucine enkephalin (*m*/*z* 554.2615), which is marked with an asterisk.

**Table 1 ijms-25-07752-t001:** Diagnosis related to surgically removed samples.

Species	Age	Gender	Diagnosis
Cats	17 years 8 months	Female	Breast simplex tubulopapillar carcinoma and its skin metastasis, grade III.
Dogs	9 years 6 months	Female	Complex breast adenocarcinoma and its lung metastasis, grade II.

## Data Availability

The data presented in this study are available on request from the corresponding author.

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
