# Peer review of "Characterisation of Canine and Feline Breast Tumours, Their Metastases, and Corresponding Primary Cell Lines Using LA-REIMS and DESI-MS Imaging"

_ijms, 2024, doi:10.3390/ijms25147752_

Round 1

Reviewer 1 Report

Comments and Suggestions for Authors

The manuscript provides insights into the feasibility of Desorption Electrospray Ionization (DESI) and Laser-Assisted Rapid Evaporative Ionization Mass Spectrometry (LA-REIMS) imaging technologies to study the various types of tumors. The authors compared the breast tumors and their metastases with their respective cell lines using these techniques and demonstrated the significant differences between tumors. However, they did not observe much difference in tissues and their respective cell lines. The microenvironment in tissues and cultured cells could be different due to their special external environment. This could potentially affect the lipid profile and the authors may observe the difference. Surprisingly, the authors did not observe such a difference. The whole idea of this manuscript is built upon differences in tumors by two different techniques which give almost similar information.  Although the manuscript is interesting and written well, it does not add much to the knowledge base.

1.     The authors analyzed different types of tumors with their respective cell lines by two MS methods. It is not clear how many molecules reproduce across the biological and technical replicates in both methods. How do the authors select the lipid molecules? The methods section needs more data acquisition details for both methods.

2.     In Figure 2, the authors analyzed the breast simplex tubulopapillar carcinoma with DESI-MS and LA-REIMS imaging. It is not clear if the same tissue section was used for both techniques. The lipid profile identified most of the similar molecules with some additional information. How many of those were already known to be associated with tumors?

3.     Figure 3 is not clear. What is on the Y-axis? What is the fold change of each of the lipid molecule?

4.     Figures 4 and 5 need to improve and make the PCA better on the right scales to capture the differences.  

5.     The differences in the metabolite mass spectra profile in Figure 7 are very informative. The authors need to label some important m/z to respective metabolites or certain regions where the peaks are similar and different. Also, why did the Breast simplex tubulopapillar carcinoma tumor tissue mass spectra start acquiring at m/z 150 and not at 70?

Minor points:

1.     The title needs to be revised.

Reviewer 2 Report

Comments and Suggestions for Authors

Although intriguing, the article proves quite challenging to read and, in my opinion, would benefit from a more fluid rewrite. Nonetheless, while I find it to be fascinating, there are several significant changes necessary before moving forward with any further revision steps.

Although intriguing, the article proves quite challenging to read and, in my opinion, would benefit from a more fluid rewrite. Nonetheless, while I find it to be fascinating, there are several significant changes necessary before moving forward with any further revision steps.

·        The text in paragraph 2.1 appears to resemble a figure caption rather than a clear paragraph. Consequently, it necessitates revision to enhance readability and clarity.

·        The readability of the entire text is hindered by the inconsistent use of abbreviations. Although they are initially introduced, authors intermittently present the full name followed by the abbreviation, solely the full text, or only the abbreviation. It is imperative to standardize the text by utilizing only abbreviations after they have been defined at the beginning of the Results and Discussion or Introduction sections. (see R 120, R 167, R 199, R 218, R207, …)

·        The article title has been truncated during editorial management due to its excessive length. I recommend removing the phrase "through Comprehensive Molecular Analyses" and the techniques' acronyms. As an alternative, I propose a more concise title:

o   "Characterization of Canine and Feline Breast Tumors, Metastases, and Corresponding Primary Cell Lines Using Ambient Ionization and Mass Spectrometry Imaging"

·        The authors identify certain lipids in different regions as either over or under-expressed based on their intensity. However, this measurement may not accurately reflect lipid concentration, as it can be influenced by factors such as matrix effects. Have the authors considered the potential for differential matrix effects between the two analyzed portions? While not a completely rigorous solution, employing a labeled standard (e.g., equisplash) uniformly distributed on both healthy and diseased cells prior to mass spectrometry analysis could strengthen the results.

·        What is m/z 564.536?

·        Furthermore, some attributions may be debatable. For instance, in negative polarity, it is known that phosphatidylcholines (PC) can ionize as demethylated adducts, isomers of phosphatidylethanolamines (PE) with two additional CH2 units on the side chains. The same applies to plasmalogen forms. Without MS/MS spectra or without comparing negative polarity data with that obtained in positive polarity, it's not possible to definitively attribute the signals obtained to PE or PE-O rather than to PC or PC-O. MS/MS spectra of main lipids could be added as supplementary material.

·        Similarly, phosphatidic acids (PA) can be generated as artifacts during the ionization process of phosphatidylserine (PS), phosphatidylinositol (PI), or phosphatidylglycerol (PG). This could explain the divergent observations obtained using the two techniques. Employing standards to validate the approach could significantly bolster confidence in the attributions.

·        Could you provide a more detailed explanation of those steps?

o   R 143: Using an unsupervised algorithm (k-Nearest Neighbours (kNN) algorithm) for clustering, (you can add also (see materials and methods section)

o   R 148 confusion matrix containing the relative frequencies of true positive (TP), true negative (TN), false positive (FP), false negative (FN) classifications was obtained, and the results were described by accuracy (ACC) and sensitivity (SEN).

·        It seems that Figures S1 and Figure 2, as well as Figure S2 and Figure 2, overlap. Perhaps it's unnecessary to include the first column from both Figures S1 and S2.

·        It would be useful to append "however, these hypotheses need to be validated" at the end of R 187.

·        Why m/z 972 and other ions are marked with “!” in Figure 7?

·        Please review the axes titles of the figures.

·        What does it means “If a spectrum was further from the class average than 15 times the standard deviation of the full class, the spectrum was labelled ‘outlier’.” What is the full class? The whole Paragraph 2.13 lacks clarity, in my opinion.

Round 2

Reviewer 1 Report

Comments and Suggestions for Authors

The authors revised the manuscript as per the comments. The manuscript now looks much improved and can be accepted.

Minor comments:

1. The title may be improved as the ambient ionization is also part of mass spectrometry. The authors may put DESI and LA-REIMS in the title. These are standard and well-known shot forms of these methods in the field.

2. Figures 4 and 5, I mean to add and label the XYZ axis instead of putting percentage in the figure caption.

Reviewer 2 Report

Comments and Suggestions for Authors Please see the attached file.
